# Evaluation of the Mammalian Aquaporin Inhibitors Auphen and Z433927330 in Treating Breast Cancer

**DOI:** 10.3390/cancers16152714

**Published:** 2024-07-30

**Authors:** Verodia Charlestin, Elijah Tan, Carlos Eduardo Arias-Matus, Junmin Wu, Maria Cristina Miranda-Vergara, Mijoon Lee, Man Wang, Dharma T. Nannapaneni, Parinda Tennakoon, Brian S. J. Blagg, Brandon L. Ashfeld, William Kaliney, Jun Li, Laurie E. Littlepage

**Affiliations:** 1Department of Chemistry and Biochemistry, University of Notre Dame, Notre Dame, IN 46556, USA; vcharles@nd.edu (V.C.); ptennako@nd.edu (P.T.); bblagg@nd.edu (B.S.J.B.); bashfeld@nd.edu (B.L.A.); 2Harper Cancer Research Institute, South Bend, IN 46617, USAjun.li@nd.edu (J.L.); 3Biotechnology Department, Life and Health Sciences Deanship, Universidad Popular Autonoma del Estado de Puebla (UPAEP University), 13 Poniente No. 1927, Barrio de Santiago, Puebla 72410, Mexico; 4Warren Family Research Center for Drug Discovery and Development, University of Notre Dame, Notre Dame, IN 46556, USA; 5Department of Applied and Computational Mathematics and Statistics, University of Notre Dame, Notre Dame, IN 46556, USA

**Keywords:** aquaporin, breast cancer, inhibitor, Auphen, Z433927330

## Abstract

**Simple Summary:**

Aquaporins (AQPs) have emerged as potential predictors of response to cancer therapy and as targets for increasing sensitivity to treatment. However, systemic therapeutic inhibition using pan-AQP or AQP-specific inhibitors has not been characterized for efficacy in treating breast cancer or as a component of combination treatment. We evaluated AQP inhibition using established AQP inhibitors in cytotoxicity and pharmacologic assays. This study identified AQPs as a targetable vulnerability in breast cancer. AQP inhibitors can increase therapeutic efficacy, both as single agents and in a combination therapy strategy, in treating breast cancer progression and metastasis. These results provide significant insights that support the future development of improved AQP inhibitors.

**Abstract:**

AQPs contribute to breast cancer progression and metastasis. We previously found that genetic inhibition of Aqp7 reduces primary tumor burden and metastasis in breast cancer. In this study, we utilized two AQP inhibitors, Auphen and Z433927330, to evaluate the efficacy of therapeutic inhibition of AQPs in breast cancer treatment. The inhibitors were evaluated in breast cancer for both cytotoxicity and metabolic stability assays across both murine and human breast cancer cell lines. Both AQP inhibitors also affected the expression of other AQP transcripts and proteins, which demonstrates compensatory regulation between AQP family members. As a single agent, Auphen treatment in vivo extended overall survival but did not impact primary or metastatic tumor burden. However, Auphen treatment made cells more responsive to chemotherapy (doxorubicin) or endocrine treatment (tamoxifen, fulvestrant). In fact, treatment with Tamoxifen reduced overall AQP7 protein expression. RNA-seq of breast cancer cells treated with Auphen identified mitochondrial metabolism genes as impacted by Auphen and may contribute to reducing mammary tumor progression, lung metastasis, and increased therapeutic efficacy of endocrine therapy in breast cancer. Interestingly, we found that Auphen and tamoxifen cooperate to reduce breast cancer cell viability, which suggests that Auphen treatment makes the cells more susceptible to Tamoxifen. Together, this study highlights AQPs as therapeutic vulnerabilities of breast cancer metastasis that are promising and should be exploited. However, the pharmacologic results suggest additional chemical refinements and optimization of AQP inhibition are needed to make these AQP inhibitors appropriate to use for therapeutic benefit in overcoming endocrine therapy resistance.

## 1. Introduction

Aquaporins (AQPs) are a family of small transmembrane proteins that facilitate the transport of water, glycerol, gases, and other small molecules across plasma cell membranes [1,2,3,4,5]. The 13 mammalian AQPs can be divided into two subfamilies: AQPs that are water-selective (AQPs 0–2, 4–6, and 8) and aquaglyceroporins that are glycerol-permeable (AQPs 3, 7, 9, and 10) [6,7]. Increasing evidence finds links between aberrant AQP expression and key biological functions in cancer, including proliferation, tumor type, grade, and prognosis [8,9,10]. In breast cancer, overexpression of AQP1, 3, 4, 5, and 7 correlates with proliferation, tumor type, grade, prognosis, and overall and relapse-free survival [8]. Downregulation and inhibition of AQP expression in breast cancer reduce pro-tumorigenic phenotypes, such as cell proliferation, tumor growth, and metastasis, making AQPs an attractive therapeutic target [8,10,11,12,13,14,15,16,17,18,19].

Previously, we discovered that Aquaporin-7 (human AQP7/mouse Aqp7), a water and glycerol channel, is a novel regulator of breast cancer [11]. We identified AQP7 as a negative prognostic marker of overall survival and metastasis in breast cancer patients. Both in vitro and in vivo experiments showed that AQP7 is required for proliferation, primary tumor progression, and metastasis. Metabolomics of Aqp7 knockdown cells and tumors revealed significantly altered lipid levels, redox, and urea/arginine metabolism. Given the correlation of AQP7 with breast cancer and its involvement in glycerol transport and lipid homeostasis, we investigated AQP7 as a metabolic target for cancer therapy. We found that AQP7 is a critical regulator of metabolic and signaling responses to environmental cellular stresses [11]. AQP7 has been associated with multiple disease states, including obesity, insulin resistance, and thyroid and breast cancers, indicating the potentially broad clinical application of an AQP7-selective modulator [11,20,21,22,23,24,25,26,27,28,29]. However, no studies have characterized the effect of AQP inhibitors, either pan-AQP or AQP7-specific inhibitors, as a component of a combination treatment to demonstrate a causal relationship between AQPs and treatment, or if combination treatment can overcome breast cancer metastasis or therapeutic resistance.

Auphen was originally identified as an AQP3 inhibitor [30]. In human erythrocytes, Auphen treatment resulted in a drastic reduction in glycerol permeability via AQP3 inhibition and had little effect on AQP1 water permeability. Auphen has also been shown to inhibit water and glycerol permeability via AQP7 in adipocytes [31]. Given the promising results seen with Auphen, new gold-based derivatives have been evaluated for their efficacy in inhibiting water and glycerol permeability. Studies of these derivatives showed a modest effect on water permeability but a significant reduction in glycerol permeability, making these derivatives effective inhibitors of glycerol transport [32]. Gold complexes such as Aubipy and Auterpy had IC50 values in the low micromolar range, which was comparable to Auphen [32]. Interestingly, metal derivatives such as Cuphen and Ptphen showed significantly decreased AQP3 inhibition, where over a 100-fold increase in IC50 was observed.

AQPs are tetrameric proteins where each monomer can function as an independent water channel. For AQP3, each monomer contains a Cys40 residue that can interact with Auphen. Auphen binds to all four of these residues corresponding to its potency and indicating cooperativity [32]. Other gold complexes have intermediate cooperativity or less binding to all four residues [32]. In AQP7, the channel has no cysteine residues; instead, methionine residues are predicted to be the binding sites of Auphen and are where glycerol transport can be disrupted [33]. Given the roles of AQPs in cellular functions like proliferation, Auphen has been tested for its ability to inhibit cell proliferation [33,34,35]. Auphen decreases cell proliferation in both cancerous and non-cancerous cell lines via AQP3 inhibition. AQP3 inhibition correlated with a strong cell cycle arrest in the S-G2/M phase in Auphen-treated cells [33]. We selected Auphen for subsequent research due to its antiproliferative effect, ability to inhibit water and glycerol transport, and low IC50.

A screen of a library of 7360 small molecules for inhibitors of mouse AQP3 using a calcein fluorescence quenching assay identified additional inhibitors [36]. DFP00173 was identified as a selective and potent AQP3 inhibitor. Surprisingly, this study also identified Z433927330 as a potent and efficacious inhibitor of mouse AQP7 water permeability. DFP00173 and Z433927330 were the most potent library compounds in inhibiting glycerol permeability and also blocked H_2_O_2_ permeability in human erythrocytes and Chinese hamster ovary cells. Given that numerous studies have shown the role of AQP3 and AQP7 overexpression in promoting breast cancer, the combined inhibition of AQP3 and AQP7 could be a useful strategy in the treatment of breast cancer progression and metastasis [11,14,15,37,38,39,40]. However, DFP00173 selectively inhibited AQP3, with only minor inhibition of AQP7 and AQP9 [36]. We focused on characterizing Z433927330, which inhibited AQP7 most potently (IC50, 0.2 µM) and inhibited AQP3 (IC50, 0.7–0.9 µM) and AQP9 (IC50, 1.1 µM) less potently.

Neither Auphen nor Z433927330 has been characterized for their efficacy in treating breast cancer. In this study, we began to evaluate Auphen and Z433927330 for their biological properties as potential therapeutics. We evaluated the systemic consequences of aquaporin inhibition and tested the efficacy of Auphen and Z433927330 as a therapeutic strategy in the treatment of breast cancer metastasis.

## 2. Materials and Methods

### 2.1. Syntheses of Auphen and Z433927330

The synthesis of Z433927330 (1) commenced from commercially available ethyl 4-aminobenzoate (2) by first treatment with phenyl choroformate to provide the corresponding carbamate 3, with an 80% yield (Figure 1) [41]. Subsequent acylation of benzyl amine 4 with carbamate 3 provided Z433927330 with a 73% yield. Synthesis of Auphen was accomplished by the treatment of phenanthroline with HAuCl_4_•3H_2_O in refluxing EtOH [42]. (see Appendix A for more details).

### 2.2. Cell Culture

All cells were maintained at 37 °C in humidified incubators with 5% CO_2_ at atmospheric oxygen levels. Mouse 4T1 (CRL-2539) and human AU565 (CRL-2351), BT474 (HTB-20), MCF-7, and MDA-MB-231 cells were purchased from ATCC. Vo-PyMT cells were a gift from Dr. Conor Lynch, Moffitt Cancer Center, University of South Florida [43]. 4T1 and AU565 cells were cultured in ATCC-formulated RPMI-1640 Medium (Thermo Scientific, Waltham, MA, USA, Cat. # A1049101) with 10% FBS. BT474 cells were cultured in Hybri-Care medium (ATCC, Cat. # 46-X), 1.5 g/L sodium bicarbonate, and 10% FBS. PyMT and MDA-MB-231 cells were cultured in DMEM High Glucose media (Sigma D5648) with 10% FBS. MCF-7 cells were cultured in DMEM High Glucose media with 10% FBS and 1 μg/mL insulin. All cells were routinely tested for mycoplasma using a colorimetric detection kit (InvivoGen rep-pt1) according to the manufacturer’s directions. Quality control of the cell lines was maintained by continual authentication of morphology and growth rate. Mouse cell lines were not authenticated genetically.

### 2.3. Drug Preparation and Treatment

All drug compounds were aliquoted as stock solutions and frozen at −20 °C. All aliquots were thawed no more than three times. Doxorubicin (Sigma, Cat. # D1515) was dissolved in either filter-sterilized ddH_2_O or DPBS at a concentration of 1 mM. For in vitro experiments, doxorubicin was used at a final concentration ranging from 0 to 20 μM, as indicated. Fulvestrant (Selleckchem, Houston, TX, USA, Cat. # S1191) was dissolved in DMSO at a concentration of 2 mM. For in vitro experiments, fulvestrant was used at a final concentration ranging from 0 to 40 μM, as indicated. Tamoxifen (Fisher Scientific, Hampton, NH, USA, Cat. # MP215673891) was dissolved in DMSO at a concentration of 1 mM. For in vitro experiments, tamoxifen was used at a final concentration ranging from 0 to 20 μM, as indicated. For in vivo experiments, tamoxifen was dissolved in 10% or 100% ethanol and 90% corn oil (Sigma Aldrich, St. Louis, MO, USA, Cat. #C8267) and heated at 37 °C for 30 min–1 h until dissolved. Tamoxifen was administered intraperitoneally at a concentration of 500 μg/mouse five times a week.

Auphen and Z433927330 were dissolved in DMSO at a concentration of 1 mM. Aliquots were stored at −20 °C for a maximum of 1 month. For in vivo experiments with PyMT cells, Auphen was dissolved in 10% DMSO and 90% corn oil. For in vivo experiments with 4T1 cells, Auphen was dissolved in 10% DMSO and 90% normal saline. Auphen was administered intraperitoneally at a concentration of 25 mg/kg either weekly or biweekly, as indicated. For in vivo experiments, Z433927330 was dissolved in 10% DMSO and 90% corn oil. Z433927330 was administered intraperitoneally at a concentration of either 15 or 25 mg/kg weekly or biweekly, as indicated.

### 2.4. Cell Viability and IC50 Determination

5000 cells were seeded in white flat bottom 96-well plates (Costar, Washington, DC, USA, Cat. # 3917). After overnight incubation, the cells were treated with Auphen or Z433927330 for 72 h at concentrations from 0–40 μM. The cells were washed with DPBS once, 100 μL of 1:1 DPBS/CellTiter-Glo 2.0 (Promega, Madison, WI, USA, Cat. # G9242) was added to each well, mixed for 2 min using a Fisher Vortex Genie 2.0, and incubated at room temperature for 10 min. Luminescence was read with a SpectraMax iD5 microplate reader. Experiments were completed using quadruplicate samples and repeated at least twice. The IC50 values were calculated via nonlinear regression using GraphPad Prism.

### 2.5. XTT Assay

HepG2 cells (ATCC HB-8065) were maintained in monolayer culture at 37 ˚C and 5% CO_2_ in Dulbecco’s Modified Eagle’s Medium (Corning, Corning, NY, USA, Cat. # 10-009-CV) supplemented with 10% fetal bovine serum, non-essential amino acids, 2 mM L-glutamine, and 1% penicillin-streptomycin. After overnight incubation, the cells were treated with Auphen or Z433927330 for 16 h at concentrations from 2 to 128 µg/mL. The cells were washed with PBS twice, and 150 µL of XTT working solution was added to each well, followed by 3 h incubation. Absorbances at 475 nm (test wavelength) and 660 nm (reference wavelength) were read with a microplate reader. Experiments were completed with quadruplicate samples, and the experiment was repeated twice. The IC50 values were calculated via nonlinear regression with GraphPad Prism.

### 2.6. Microsomal Stability

The drugs were incubated with liver microsomes and analyzed using UPLC/MS to evaluate their liver half-life stability. Incubations consisted of male rat pooled liver S9 (Corning Gentest, Cat. # 452591, 0.5 mg/mL, final concentration), NADPH (Cayman Chemical Company, Ann Arbor, MI, USA, Cat. # 2646-71-1, 0.5 mM, final concentration), and 50 μM of the indicated compound (final concentration) in potassium phosphate buffer (50 mM, pH 7.4) at 37 °C in a total volume of 1 mL. Aliquots were drawn at different time points, and the reactions were terminated by the addition of one volume of acetonitrile containing 1 μM of an internal standard (SB225002, MedChemExpress, Monmouth Junction, NJ, USA, Cat. # HY-16711/CS-3538). The precipitated protein was centrifuged at 20,000× *g* for 30 min. A solid phase extraction (SPE) clean-up protocol was conducted on the supernatant using Oasis HLB 1cc (30 mg) extraction cartridges (Oasis, Woburn, MA, USA, Part # WAT094225). Columns were conditioned and equilibrated with 1 mL of MeOH and 1 mL of H_2_O, respectively. Supernatants were then loaded onto columns. Columns were washed with 1 mL of 95:5 water/acetonitrile and then eluted with 1 mL of MeOH. The final eluted solution was diluted with an equal volume of water and analyzed using UPLC/MS. Peak areas from extracted-ion chromatograms of corresponding *m*/*z* value (365.16 for the Z433927330 compound, 351.99 for the internal standard) were measured, and peak-area ratios of the compound compared to the internal standard were used to calculate half-lives of the compound in human and rat liver microsomes.

The UPLC/MS instrument consisted of a Waters Acquity H-Class UPLC equipped with a Waters Acquity sample manager-FTN and PDA detector coupled with a Bruker impact II ultrahigh resolution Qq-time-of flight hybrid mass spectrometer controlled by Bruker Compass HyStar version 5.0 SR1. The Bruker electrospray ionization source was operated in the positive ion mode with the following parameters: end plate offset voltage = −500 V, capillary voltage = 1800 V, and nitrogen as both a nebulizer (4 bar) and dry gas (7 L/min) at 200 °C. Mass spectra were accumulated over the mass range of 150–3000 *m*/*z*. LC separations were generated on an Acquity UPLC BEH C18 column (Waters; 2.1 × 50 mm, 1.7 μm) connected to an Acquity UPLC BEH C18 VanGuard^TM^ Pre-column (Waters; 2.1 × 5 mm, 1.7 μm) at 40 °C. A 10 min LC gradient was performed as follows: hold at 30% B for 1 min, ramp to 90% B for 7.4 min, ramp to 30% B for 0.1 min, and hold at 30% B for 1.5 min (A = 0.1% formic acid in water, B = 0.1% formic acid in acetonitrile) at a flow rate of 0.4 mL/min. LC flow during the first 1 min was diverted to the waste.

### 2.7. RNA Extraction and Reverse Transcription Quantitative Polymerase Chain Reaction (RT-qPCR)

RNA extraction was carried out with RNA-STAT 60 (Amsbio, Cat. # CS-110) following the manufacturer’s instructions. Cultured cells were scraped from culture dishes and washed with PBS before flash freezing with liquid nitrogen. Cells and tissue were homogenized with appropriate amounts of RNA-STAT 60. Phase separation was carried out by adding 0.2 mL chloroform per 1 mL of RNA-STAT 60, mixing, and centrifuging at 12,000× *g* for 15 min. RNA from the top aqueous layer was precipitated with isopropanol, washed with 75% ethanol, dried, and dissolved in RNase-free water. The RNA concentration and quality were measured with a NanoDrop 2000 spectrometer (260/280 > 2.0).

cDNA synthesis from RNA was carried out using the QuantiTect Reverse Transcription Kit (Qiagen, Hilden, Germany, Cat. # 205311) following the manufacturer’s protocol. Briefly, 1 µg of RNA sample was aliquoted, and genomic DNA was removed using the included gDNA Wipeout Buffer. Reverse transcription was then carried out in a 20 µL final volume, with RT primer mix and reverse transcriptase. cDNA was diluted to 400 µL, and 2–4 µL was used for each qPCR reaction.

RT-qPCR was carried out using 2× SYBR Green qPCR Master Mix (Bimake, Cat. # B21203) according to the manufacturer’s recommendations. Primers for qPCR were ordered from Sigma-Aldrich (KiCqStart^®^ SYBR^®^ Green Primers) (see Appendix A for the full list of primer sequences).

qPCR was carried out in a 20 µL reaction in MicroAmp Optical 96-well Reaction Plates (Applied Biosystems, Waltham, MA, USA, Cat. # N8010560) with a QuantStudio 3 Real-Time PCR. No template and no primer controls were included with each run, and PCR products were analyzed with melting curves as well as with 2% agarose gels to confirm amplification. Experiments for each gene were replicated more than three times using both biological and technical triplicates. * *p* < 0.05; **, *p* < 0.01; ***, *p* < 0.001, unpaired *t* test.

### 2.8. Western Blot

Cell pellets were collected after 3 days of the indicated treatment. Protein lysates were generated using RIPA lysis buffer (150 mM sodium chloride, 1% NP-40 or Triton X-100, 0.5% sodium deoxycholate, 0.1% SDS, 50 mM Tris (pH 8), 5 mM EDTA, and 0.5% CHAPS) containing freshly added 1 mM PMSF (Sigma, Cat. # 11359061001, 250 mM stock), 1× cOmplete Protease Inhibitor Cocktail (Roche, Cat. #11697498001, 25× stock), and 1× PhosStop Phosphatase Inhibitor (Roche, Basel, Switzerland, Cat. # 4906845001, 10× stock). Protein lysate samples were quantified using the DC Protein Assay (Bio-Rad, Hercules, CA, USA, Cat. # 5000116). A total of 15–30 µg of protein per sample was combined with sample buffer, boiled for 5 min, and loaded and separated by electrophoresis on 12% SDS polyacrylamide gels. The proteins were transferred to nitrocellulose membranes (Bio-Rad, Cat. #162-0094) and blocked with 5% BSA in PBST. The membrane was stained with primary antibody overnight at 4 °C, followed by incubation with HRP-conjugated secondary antibody (Cytiva, Marlborough, MA, USA, Cat. # NA931V (mouse) or NA934V (rabbit)). Blots were visualized with Clarity Western ECL Substrate (Bio-Rad, Cat. #170-5061) and imaged using Bio-Rad ChemiDoc MP. All western samples were compared as biological triplicates. The following antibodies were used: AQP3 (F-1) (Santa Cruz, Cat. # sc-518001) (1:200 of 200 µg/mL stock solution), AQP7 (D-12) (Santa Cruz, Cat. # sc-376407) (1:200 of 200 µg/mL stock solution), AQP9 (G-3) (Santa Cruz, Cat. # sc-74409) (1:200 of 200 µg/mL stock solution), Aqp7 (Millipore Sigma, Cat. # AB15568) (1:1500 of 0.8 mg/mL stock solution), and β-Actin (13E5) (Cell Signaling, Danvers, MA, USA, Cat. # 4970S) (1:3000 of 61 µg/mL stock solution).

### 2.9. Animals

Mice used in this study were maintained under pathogen-free conditions at the University of Notre Dame Freimann Life Sciences animal facility. Animal protocols were approved by the University of Notre Dame Institution Animal Care and Use Committee. Animal experiments were conducted in accordance with the approved protocol guidelines (21-11-6912, 21-12-6936).

#### 2.9.1. Orthotopic Injections

BALB/cJ, NOD SCID, or FVB/N mice from eight to ten weeks of age were used as recipient mice for transplantation surgeries. Prior to injections, mouse 4T1 or PyMT mammary cancer cells are cultured, trypsinized, washed, and resuspended in PBS at a concentration of 1.25 × 10^5^ cells/20 µL or 1 × 10^5^ cells/20 µL, respectively, and injected into the fourth mammary glands. Mice were monitored daily for weight, and, once tumors were palpable, tumor size was measured three times a week with a digital caliper. Tumor volumes were calculated using the formula tumor volume = πLW^2^/6, with the width being the smaller of the two measurements. The mice were sacrificed prior to the biggest tumor reaching 2.0 cm in diameter, as required by IACUC. As an alternative, mice were collected as tumors reached 2.0 cm in diameter to calculate overall survival. The tumors were weighed, cut into several pieces, and flash-frozen or fixed in 4% PFA. The lungs from these mice were also collected, visible metastases were counted, and the lungs were fixed in 4% PFA and embedded in paraffin for histological analysis. Micrometastases were quantified and blinded by pathological analysis of H&E sections of the lungs by a trained pathologist to confirm results. Animals that had no measurable tumors prior to treatment were excluded from the analysis.

#### 2.9.2. Statistical Analysis

For comparison of tumor volume, we assumed that each tumor grew exponentially over time. For the *j*’th measurement of mouse *i*, which is done on day *X_ij_*, we modeled its tumor measurement *Y_ij_* using log *Y_ij_* = α_i_ + (β_1_ + β_2_*Z_i_*)*X_ij_*. Here, *α_i_* takes care of the initial measurement of the tumor on day 0, which is different for each mouse. *Z_i_* is an indicator function, which equals 1 if the sample belongs to cohort 2 (e.g., case) and 0 if the sample belongs to cohort 1 (e.g., control). The growth rates of the two cohorts are β_1_ and β_1_ + β_2_, respectively. To test whether the growth rates were different between the two cohorts, we tested whether or not β_2_ is 0. A *p*-value was given by the model. A negative β_2_ value indicated that the growth rate was lower in cohort 2. When a *p*-value was significant, we checked whether the value of β2, which is shown in the “Estimate” column, was positive or negative. A positive value means that Cohort 2 grows faster than Cohort 1, and a negative value means that Cohort 2 grows slower than Cohort 1.

For comparison of lung metastasis, the data were non-negative integers and, thus, treated as counts, and a Poisson log-linear model was used. When the results were statistically significant (i.e., Pr(>|z|) < 0.05), the two cohorts had significantly different numbers of metastasis. In this case, a positive “Estimate” value means that Cohort 2 has a larger number of metastasis than Cohort 1, and a negative “Estimate” value means that Cohort 2 has a smaller number of metastasis than Cohort 1.

To compare overall survival, a log-rank test was used for the comparison of survival time between each pair of groups.

### 2.10. Immunohistochemistry

Sectioned tumor tissues on slides were deparaffinized in an oven at 60 °C for 1 h, washed three times in xylene, and rehydrated in decreasing concentrations of ethanol five times. Sodium citrate antigen retrieval was completed by submerging the slides in 10 nM sodium citrate solution, followed by boiling the slides with the sodium citrate solution for 7 min, two times at 70% power in a microwave. After cooling for 35 min, the slides were loaded into a humidified chamber and washed three times with PBS. Slides were then incubated in freshly prepared 3% H_2_O_2_ (Fisher Scientific, Cat. # H325-100) in PBS for 5 min and washed three times with PBS and blocked using the Avidin-Biotin Blocking kit (Vectastain Elite, Cat. # PK-6100) following the manufacturer’s directions. Slides were then blocked with 20% goat serum (Jackson ImmunoResearch Laboratories, Inc., West Grove, PA, USA) in PBS for 30 min and left overnight in the primary antibody in PBS at 4 °C. The next day, slides were incubated at room temperature for 30 min and washed three times with PBS. The antibodies recognizing Ki67 (Cell Signaling Technology, Danvers, MA, USA, Cat. # 12202T, 1:400 of 270 µg/mL stock solution), Cleaved caspase 3 (Cell Signaling Technology, Cat. # 9661, 1:300 of 52 µg/mL stock solution), and the biotinylated goat anti-rabbit IgG (H + L) secondary antibody (Vector Laboratories, Cat. # BA-1000) (1:250 dilution with PBS) were added for 30 min. Slides were washed three times with PBS, incubated with the ABC solution (Vector Laboratories, Newark, CA, USA, Cat. #PK-6100) for 30 min, washed 3 times, and developed with ImmPact DAB (Vector Laboratories, Cat. # SK-4105) in PBS. Slides were counterstained in Hematoxylin QS (Vector Laboratories, Cat. # H-3404) for 1 min, washed in RO water for 5 min, and submerged in increasing concentrations of ethanol and dehydrated in xylene. Slides were then mounted with Cytoseal XYL and covered with #1.5 coverslips (Thermo Scientific, Cat. #22-050-244, 24 × 50 mm) and dried.

Slides were scanned using an Aperio CT scanner (Aperio Technologies, Vista, CA, USA) with a 20× objective. Digital images were saved on the eSlide Manager database. The digitized images for the indicated markers used were annotated/analyzed in the most representative area, strictly in the tumorigenic epithelia excluding the stromal-rich and necrotic regions, at the interface of the stromal/epithelial regions or stromal regions specifically. Analysis of the digitalized IHC images of tissues stained for Ki67 and cleaved caspase-3 used a developed Aperio nuclear or cytoplasmic algorithm specific for each marker to score DAB and hematoxylin chromogen intensities. Data generated from the analysis was exported from ImageScope to an Excel spreadsheet for statistical analysis.

### 2.11. RNA-Seq

Cell pellets were collected after 3 days of Auphen treatment. RNA extraction was carried out with RNeasy Plus Mini Kit (Qiagen, Cat. # 74136) following the manufacturer’s protocol. Total RNA was evaluated with Agilent Tapestation 4150 system and RNA ScreenTape (Agilent Technologies, Santa Clara, CA, USA). All samples had an RNA Integrity Number (RIN) of 9 or higher. Samples were quantitated with the Qubit RNA HS Assay Kit (Invitrogen, Carlsbad, CA, USA).

Total RNA input was normalized to 500 ng. mRNA was enriched using NEBNext Poly(A) mRNA Magnetic Isolation Module (PN: E7490L; New England BioLabs, Ipswich, MA, USA) and converted into an Illumina library using NEBNext Ultra II Directional RNA Library Prep with Sample Purification Beads (PN: E77765L; New England BioLabs, Ipswich, MA, USA). Indexed libraries were quality assessed using Qubit dsDNA HS Assay Kit (PN: Q32854, Invitrogen, Carlsbad, CA, USA) and Tapestation 4150 DNA HS 1000 Kit (Agilent Technologies, Santa Clara, CA, USA). The individual libraries were normalized, and equal molar amounts were multiplexed. The molar concentration of the multiplex pool was determined using KAPA Library Quantification Kits for Illumina (PN: KK4824; KAPA Biosystems, Boston, MA, USA). Libraries were sequenced on one lane of a NovaSeq S4 (300 cycles) flowcell (Illumina Inc., San Diego, CA, USA) at the Indiana University Center for Medical Genomics using paired 150 bp reads with a target of 50 M reads (clusters) per sample.

Reads were mapped to human (genome GRCh38, transcriptome GRCh38.109) or mouse genomes (genome GRCm39, transcriptome GRCm39.109) using HISAT2 [44], and then the number of reads uniquely mapped to each gene tallied using HTSeq [45]. The percentages of reads that can be uniquely mapped (for both human and mouse) are around 65%, and all are above 50%. When identifying differentially expressed (DE) genes, vehicle and Auphen-treated samples were paired. Next, we conducted differential expression (DE) analysis using edgeR [46] using the likelihood ratio test (LRT). Low-expression genes whose average number of reads per sample < 2.0 were filtered out, except for Aqp7/AQP7. The columns for each file are GeneSymbol (the gene symbol), FDR (the false discovery rate or the adjusted *p*-value; set to 0.1 as a cutoff), pval (the unadjusted/original *p*-value), log2FC (log2 fold change. A positive/negative log2FC means that the gene is upregulated/downregulated in the second condition compared to the first condition.), and log2CPM (log2 of the overall average expression, measured by log2 of counts per million. This number indicates whether the gene is a high-expression or low-expression gene, on average, over the two groups). For candidate genes with an FDR < 0.1 and a *p*-value < 0.05, gene ontology was analyzed using DAVID (https://david.ncifcrf.gov/home.jsp, accessed on 26 July 2023, DAVID Knowlegebase (v2023q4), DAVID 2021 (Dec. 2021) [47,48]. Upregulated and downregulated genes were subjected separately to gene ontology (GO) under the GOTERM_BP_DIRECT to find significantly enriched genes and associated pathways.

## 3. Results

### 3.1. In Vitro Cytotoxicity and Metabolic Stability

To investigate the cytotoxicity of the AQP inhibitors Auphen (pan-AQP inhibitor) and Z433927330 (AQP7 inhibitor) (Figure 1A), a panel of mouse and human breast cancer cell lines were cultured and treated with different concentrations of Auphen or Z433927330, and the cytotoxic effects were assessed using CellTiter-Glo assay (Figure 1B,C). The panel of cell lines included two tumorigenic and metastatic mouse mammary epithelial cells derived from luminal B subtype MMTV-polyoma middle T antigen PyMT and triple-negative breast cancer 4T1 cells. The panel of human cell lines, including MCF-7, BT474, AU565, MDA-MB-231, and MDA-MB-468, were selected because each represents a different breast cancer subtype of Luminal A, Luminal B, HER2+, claudin-low, and basal, respectively. After 72 h of Auphen treatment, the survival rate of cell lines decreased in a dose-dependent manner and was used to calculate Auphen IC50 values for each cell line.

The Auphen IC50 values that were calculated from both mouse and human cell lines ranged from 3.7 to 8.6 µM. In murine breast cancer cells, the 4T1 cells showed a higher sensitivity to Auphen treatment than the PyMT cells (Figure 1B). In human breast cancer cells, MCF-7 and MDA-MB-231 had similar sensitivities to Auphen, with IC50 values around 3.7 (Figure 1C). AU565 and BT474 had IC50 values around 8.3–8.6 µM, indicating a reduced sensitivity to Auphen treatment compared to MCF-7 and MDA-MB-231.

The Z433927330 IC50 values from mouse and human breast cancer cell lines ranged from 11.8 to 25 µM, indicating reduced cytotoxicity compared to Auphen. Similar to Auphen treatment, 4T1 (IC50 = 16.6 µM) cells were more sensitive to Z433927330 than were PyMT cells (IC50 = 25 µM) (Figure 1B). Of the human breast cancer cells, AU565 cells were the most sensitive to Z433927330 (Figure 1C).

Potential liver cytotoxicity of AQP inhibitors was evaluated using an XTT assay following treatment of HepG2 cells with Auphen or Z433927330 (Figure 1D). The resulting IC50 values showed that Z433927330 had higher toxicity and lower IC50 values (IC50 = 60 µM) than Auphen (IC50 = 121 µM) in HepG2 cells.

We next evaluated the metabolic stability, or rate of disappearance, of Z433927330 over time in both rat and human liver S9 microsomes (Figure 1E). After 1 h of incubation, approximately 68% and 73% of Z433927330 were metabolized in rat and human liver microsomes, respectively. Unfortunately, the metabolic stability of Auphen could not be calculated using the same method because the gold complex was trapped in the SPE columns used to clean up the samples prior to running them on the LC/MS.

### 3.2. AQP Compensation after AQP Inhibition

Auphen is a pan-AQP inhibitor while Z433927330 is a potent and selective inhibitor of AQP7 that also inhibits AQP3 and AQP9. Since both inhibitors may have off-target effects or develop compensation between AQP family members, we evaluated the effects of these inhibitors on AQP3 and AQP9 expression (Figure 2). We also evaluated if therapeutic inhibition could lead to AQP redundancy or compensation by other AQPs. AQP1, AQP3, AQP4, AQP5, AQP7, and AQP9 expression were evaluated because these AQPs are expressed in breast tumors and have begun to be characterized as having roles in breast cancer progression and metastasis [11,12,13,14,15,16,18,37,49,50,51].

AQP expression was quantified using RT-qPCR of Auphen- and Z433927330-treated breast and murine cancer cell lines (Figure 2A). Auphen and Z433927330 treatment had less compensatory AQP expression in murine than in human cancer cell lines. In PyMT cells, only Aqp5 expression decreased after Auphen treatment compared to control. In 4T1 cells, Aqp1 expression increased and Aqp7 decreased after Auphen treatment compared to the control. In human breast cancer cells, we observed significant expression changes in AQP expression after Auphen treatment in each cell line and after Z433927330 treatment in all cell lines except BT474. However, the changes in expression were inconsistent (increased/decreased for each AQP) across all four cell lines tested.

We next analyzed the protein expression of AQP3, AQP7, and AQP9 in response to increased doses of Auphen or Z433927330 (Figure 2B,C). Treatment using either AQP inhibitor impacted the expression of these proteins across murine and human cell lines. In most cases, AQP proteins decreased after treatment with either Auphen or Z433927330. For example, after Auphen or Z433927330 treatment, AQP7 decreased in five of the six cell lines tested. AQP3 and 9 protein expression levels were inconsistent across cell lines (decreased in PyMT, 4T1, MDA-MB-231; increased in MCF7, BT474; stayed the same in AU565 cells). After Z433927330 treatment, AQP3, 7, and 9 protein levels decreased in all murine and human cells tested, except in 4T1 cells, where AQP7 and 9 levels did not change.

These results demonstrate that AQP inhibition may impact AQP expression within breast cancer cell lines, which can lead to compensation by other AQPs. However, the expression patterns vary by cell line and by treatment with each AQP inhibitor.

### 3.3. In Vivo Evaluation

We next assessed the impact of Auphen in vivo on mammary tumor growth, lung metastasis, and overall survival (Figure 3). 4T1 cells were injected into the mammary fat pad of NOD-SCID and Balbc/J mice (Figure 3A,E). Once tumors were palpable, the mice received intraperitoneal (IP) injections of Auphen (25 mg/kg biweekly) or vehicle control. Although we did not observe a decrease in tumor growth or lung metastasis, we observed a significant increase in overall survival in both NOD SCID and BALB/cJ mice (Figure 3B–D,F–H).

We then assessed the effect of Z433927330 on tumor growth and lung metastasis (Figure 4). 4T1 cells were injected into the mammary fat pad of syngeneic BALB/cJ mice (Figure 4A). Once tumors were palpable, the mice were treated with Z433927330 (15 or 25 mg/kg) or vehicle control. Given that no in vivo data were available for Z433927330, we initially treated the animals biweekly with Z433927330 and then repeated the experiment with weekly treatment. Like Auphen-treated mice, we observed no difference in tumor growth in either biweekly or weekly treated mice (Figure 4B,E). We also analyzed lung metastasis and saw decreased lung metastasis after weekly treatment with Z433927330 but not in the biweekly treatment cohort (Figure 4C,F). Unlike the Auphen treatment, we observed no difference in overall survival in either weekly or biweekly treated mice (Figure 4D,G).

### 3.4. In Vitro Combination Treatment

We previously evaluated the ability of AQPs to be predictive of patient outcomes in breast cancer using ROC plotter, an online transcriptome-level validation tool for predictive biomarkers [8,52]. ROC plot analysis identified AQP7 as a potential cancer biomarker. Patients with tumors that responded to any endocrine therapy had reduced AQP7 expression for Luminal A subtype breast cancer. Therefore, AQP7 expression was higher in tumors from breast cancer patients that did not respond to endocrine therapy than it was compared to those tumors that did respond.

Because of this connection between AQP7 expression and endocrine therapy response, we investigated if AQP inhibition improved the therapeutic efficacy of endocrine therapy in breast cancer in vitro (Figure 5). Auphen cooperated with doxorubicin to reduce the viability of breast cancer cells, which suggests that Auphen treatment makes the cells more susceptible to doxorubicin (Figure 5A). AQP inhibition also cooperatively elicits antitumorigenic responses in combination with tamoxifen and fulvestrant in murine ER+ PyMT and human ER+/HER2− MCF7 breast cancer cell lines (Figure 5B,C). These results demonstrate that AQP inhibition increases the antitumorigenic effects of breast cancer therapy in multiple breast cancer models.

### 3.5. In Vivo Combination Treatment

We next asked whether combination treatment with Auphen and tamoxifen would reduce cancer progression and metastasis and increase overall survival (Figure 6). PyMT cells were injected into the mammary fat pads of FVB mice, and mice were treated with Tamoxifen and Auphen alone or in combination with Auphen and Tamoxifen (Figure 6A). Five days post-injection, mice received IP injections of either tamoxifen or corn oil (vehicle control) for five days a week. Auphen was administered by IP injections at 25 mg/kg once a week. Auphen treatment was sufficient to reduce primary tumor burden in the PyMT model, which was in contrast to that seen in the 4T1 tumors. However, Auphen (A) was not as effective as Tamoxifen alone (T) (Figure 6B).

We also tested whether the treatment order influenced cancer progression and survival using a combination treatment with Auphen and tamoxifen. Mice were initially treated with either Auphen or tamoxifen and, after two days, started receiving the other treatment. Tumor growth rates reduced significantly in response to monotherapy and combination treatment, with the slowest growth rates in the Auphen + Tamoxifen (AT) group compared to the vehicle (V) (Figure 6B). Tamoxifen-treated mice had significantly reduced growth rates compared to Auphen-only and Tamoxifen + Auphen (TA)-treated mice. AT-treated mice had slower growth rates than the vehicle but were not statistically significant compared to tamoxifen alone.

Overall survival significantly increased with monotherapy and combination treatment compared to vehicle, with the AT group having the greatest significance (Figure 6C). Auphen treatment increased the overall survival of the animals. However, the AT combination treatment was not better than tamoxifen alone, and the TA treatment was worse than tamoxifen alone. Additionally, AT treatment approached significance compared to Auphen treatment. Consistently, overall survival increased in AT-treated mice compared to TA-treated mice, indicating that treatment order is important. Therefore, the order of treatment seems to impact response, with AT decreasing tumor burden more than TA. However, the difference with AT improved over tamoxifen alone.

Lung metastasis significantly increased in Auphen-treated mice but not in tamoxifen-treated mice compared to vehicle (Figure 6D). Lung metastasis also decreased significantly with combination treatment, regardless of treatment order, compared to vehicle. In fact, combination treatment approached significance compared to tamoxifen alone. Coupled with our in vitro studies, these data suggest that AQP inhibition sensitizes breast cancer to tamoxifen.

To further evaluate the underlying cooperative effects of Auphen and tamoxifen in reducing tumor growth and metastasis, we compared tumor proliferation and apoptosis via immunohistochemistry assay for Ki67 and cleaved caspase-3 (Figure 7A,B). Neither Ki67 nor cleaved caspase-3 staining levels differed between vehicle and treatment groups. These observations suggest that Auphen treatment cooperates with tamoxifen to reduce tumor burden through an alternative mechanism.

### 3.6. RNA-Seq

To understand how Auphen contributes to the cooperative effects of combination treatment and cell viability, we analyzed gene expression by conducting RNA-sequencing (RNA-seq) analysis of vehicle or Auphen-treated PyMT and MCF-7 cells (FDR < 0.1). In treated MCF-7 cells, we identified over 15,000 genes differentially expressed between vehicle and Auphen-treated cells (Figure 8A, Appendix A). Gene ontology (GO) term analysis showed that transcriptional responses related to cell cycle, DNA repair, and mitochondrial translation were in the top 20 biological processes downregulated with Auphen treatment (Figure 8B, Appendix A), while pathways related to cell death, response to hypoxia, and transcription were upregulated. For treated PyMT cells, we identified over 14,000 genes differentially expressed between vehicle and Auphen-treated cells (Figure 8C, Appendix A). GO term analysis showed that metabolic processes, like lipid metabolism and mitochondrial function, were among the top 20 biological processes downregulated by Auphen treatment (Figure 8D, Appendix A), while changes in pathways associated with cell death, DNA damage, and transcription were upregulated.

To confirm the RNA-seq analysis, RT-qPCR analysis validated the genes of interest (Figure 8E). Of the 14 genes tested, gene expression of all but one gene (Tfam) was significant and all but three genes (Sgpp1, Ucp2, and Pgls) changed in a direction that was consistent with the RNA-seq analysis.

## 4. Discussion

AQPs play an important role in cellular functions associated with breast cancer progression, such as cell proliferation, cell migration, tumor growth, and metastasis [8,11,12,13,14,15,16,17,18]. In this study, we studied the pharmacologic effects of aquaporin inhibition and examined the consequences of these inhibitors on the viability of breast cancer cell lines in culture. We also evaluated the in vivo efficacy of Auphen and Z433927330 as therapeutic approaches in the treatment of breast cancer progression and metastasis. Treatment with either Auphen or Z433927330 affected the expression of other aquaporins, suggesting some off-target effects and compensation by other aquaporins that vary across cell lines and treatments. Auphen was more efficacious than Z433927330 both in vitro and in vivo. In addition, combination treatment of Auphen with either doxorubicin or endocrine therapy increased the cytotoxicity of breast cancer cells in cell culture and decreased tumor burden in vivo. In fact, combination treatment reduced both the tumor growth rate and metastasis and increased overall survival compared to the vehicle. However, compared to tamoxifen, the order of treatment impacted the tumor growth rate but did not significantly impact metastasis and overall survival. Finally, transcriptomic analysis of gene expression after Auphen treatment revealed significant changes in pathways related to cell cycle/division, response to DNA damage, and cell death. This study supports AQPs as therapeutic targets of breast cancer and highlights the need for improved AQP inhibitors with improved pharmacokinetic properties to properly examine their therapeutic potential.

AQPs play an important role in cellular functions associated with cancer progression, such as cell proliferation, cell differentiation, cell migration, and cell adhesion. Various studies have investigated the significance of AQPs in breast cancer. AQP1 and AQP5 overexpression plays a key role in cancer cell proliferation, migration, invasion, and chemosensitivity in several breast cancer subtypes [12,13,17,53,54,55]. Similarly, AQP3 is required for cell proliferation, migration, and invasion [14,15,38]. We previously identified AQP7 as a negative prognostic marker of overall survival and metastasis in breast cancer patients [11]. Aqp7 knockdown significantly reduced primary tumor burden and metastasis. Consistent with these Aqp7 results, in this study we discovered that therapeutic inhibition of AQP by Auphen treatment reduced the tumor growth and increased overall survival. However, surprisingly, we found that therapeutic inhibition of AQP increased lung metastasis when given as a monotherapy. Since systemic treatment would target both the cancer cells and the surrounding microenvironment, these unexpected results could suggest different roles for AQPs in heterogeneous tissue or in the in vivo context and should be the focus of future studies.

Interestingly, when Auphen was given as part of a combination treatment strategy, we observed significantly decreased cell viability, tumor growth, and metastasis and increased overall survival. Our results contribute to the accumulating evidence that AQPs can affect chemosensitivity [53,54,56,57,58,59]. Inhibition of AQP1 in breast cancer increased sensitivity to anthracycline treatment [53]. Moreover, AQP5 inhibition not only reversed Adriamycin resistance of breast cancer cells but reduced the IC50 of Adriamycin in MCF-7 cells [54]. The changes in RNA expression related to cell cycle/division, response to DNA damage, and cell death could be contributing factors that help to promote cooperativity between Auphen and tamoxifen. Future studies are required to determine the mechanisms of chemosensitivity in breast cancer in relation to AQP expression and if the changes noted above play a role.

Our study highlights the poor pharmacologic properties of Auphen and Z433927330 and their impact on multiple AQPs. Both compounds are water-insoluble and are not AQP-specific. Our pharmacological studies on both inhibitors to determine the IC50s indicate that Auphen is more potent and less toxic than Z433927330. Future research into the optimization and pharmacological analysis of AQP inhibitors will be required to increase the solubility of AQP inhibitors and to increase the selectivity of these inhibitors against specific AQPs.

Structural and molecular analyses of these inhibitors with AQP7 highlight potential differences in their mechanisms of inhibition. For Auphen, non-covalent molecular docking studies predicted that the gold compound interacts with Met47 and Met93 in the NPA region on the intracellular side of human AQP7 [33]. Cryo-EM analysis found that Z433927330 binds near the ar/R region of human AQP7, near the cytoplasmic side of the pore, and forms hydrogen bonds with loop B and Gln183 of transmembrane segment 4 [60]. These data together demonstrate that Z433927330 and Auphen have different binding sites in AQP7.

A recent study examined the treatment efficacy of Auphen in hepatocellular carcinoma, where tumor growth rates and size decreased significantly in Auphen-treated mice compared to control mice [61]. Similar to these results, after optimizing Auphen treatment with weekly dosing, we found that weekly Auphen treatment decreased tumor growth rates significantly and increased overall survival. Biweekly administration also increased overall survival but was insufficient to change tumor growth or lung metastasis.

Ours is the first study on Z433927330 efficacy in vivo. Z433927330 was originally identified as a potent and selective AQP7 inhibitor that was less specific at inhibiting glycerol permeability of AQP3 or 9 compared to DFP00173 in vitro [36]. In contrast, our results for Z433927330 treatment of breast cancer cells demonstrate that Z433927330 impacts the expression of other AQPs. Significantly, we found that weekly Z433927330 treatment decreased lung metastasis but had no effect on primary tumor growth rates or overall survival.

Given the numerous roles attributed to AQPs in tissues, a pan-AQP inhibitor may not be suitable given the potential harmful off-target effects. AQPs are widely distributed throughout the body and expressed in various epithelia and endothelial cells. These distributions include kidney tubules, brain glial cells, adipocytes, pancreas, and epidermis [62]. Knockout (KO) mice studies showed the implications of AQP3 and AQP7 KO, which led to polyuria and adipocyte hypertrophy, respectively.

Moreover, we are only beginning to understand the role of AQPs in the tumor microenvironment. The increased lung metastasis with Auphen treatment as a monotherapy may be due to off-target effects or compensation by other AQPs that contribute to cancer progression. AQP expression is necessary for the activation, trafficking, proliferation, and regulation of metabolic pathways in a number of cells that include stromal cells such as immune cells, fibroblasts, and adipocytes [8]. In the context of breast cancer, utilizing a pan AQP inhibitor that causes systemic inhibition could lead to a number of off-target effects. Studies that investigate the role of AQPs in stromal cells will be essential to determine whether there is a clinical rationale for using a general systemic aquaporin inhibitor or an epithelial-targeted aquaporin inhibitor for efficacy in treating metastatic breast cancer. Finding differences in the consequences of AQP inhibition on epithelial and stromal responses would provide a rationale that using a pan-AQP inhibitor alone may not be suitable and may act differently in vivo. In addition, future studies on AQP redundancy and compensation as mechanisms used to overcome aberrant AQP function are still needed.

## 5. Conclusions

The AQP inhibitors Auphen and Z433927330 had not yet been tested for their efficacy in treating breast cancer. Here, we characterized their effect as a component of combination treatment and tested their efficacy on breast cancer progression and metastasis. We provided significant insights to support the future development of improved AQP inhibitors that will be necessary prior to their use in clinical trials of breast cancer patients with tumors that express AQPs.

## Data Availability

All data generated or analyzed during this study are included in this published article and Appendix A. The RNA-seq data generated during this study have been deposited in the NCBI Gene Expression Omnibus (GEO) under accession number GSE271926.

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
