# Peer review of "Evaluation of the Mammalian Aquaporin Inhibitors Auphen and Z433927330 in Treating Breast Cancer"

_cancers, 2024, doi:10.3390/cancers16152714_

Round 1

Reviewer 1 Report

Comments and Suggestions for Authors

Thank you for the opportunity to evaluate your research. The work includes a lot of experiments accompanied with thorough in vivo research. 

I have observed a lack of issues regarding English.

In my opinion, using the 4T1 model in NOD-SCID mice, seems a bit excessive (section 3.3).  

I think the drug combination experiment (3.4) needs clarifications - currently there are no clear toxicity levels and combination indexes (antagonism, synergism or additive effect) presented; the data should be presented in a more clear way.

I found this study really interesting - AQPs as the novel targets for anticancer therapy, especially in terms of drug combinational application. The authors performed really huge experimental work. 

Author Response

Thank you for the opportunity to evaluate your research. The work includes a lot of experiments accompanied with thorough in vivo research. 

I have observed a lack of issues regarding English.

In my opinion, using the 4T1 model in NOD-SCID mice, seems a bit excessive (section 3.3).  

I think the drug combination experiment (3.4) needs clarifications - currently there are no clear toxicity levels and combination indexes (antagonism, synergism or additive effect) presented; the data should be presented in a more clear way.

Response: This is a great idea and will be the focus of future directions for this study. This will require extensive additional experimentation and is beyond the scope of this article.

I found this study really interesting - AQPs as the novel targets for anticancer therapy, especially in terms of drug combinational application. The authors performed really huge experimental work. 

Reviewer 2 Report

Comments and Suggestions for Authors

Authors present their work on aquaporin inhibitors in treating breast cancer. This is an interesting, novel field of anticancer research and as such it is of interest to reader of Cancer.

 Methodologically, article is a standard work in this field, and as such appropriate. Conclusions are limited at this stage, authors clearly state the limitations of their study.

 Authors extensively studied the synergistic effects of Z433927330 and several anticancer drugs, this results will be of ineptest to all researchers in the field of anticancer drugs.

 My main concern is the selectivity profile of Z433927330, too often compound selectivity is not evaluated rigorously enough, therefore biological results are not so uniform as excepted.

Also authors have some issues with this topic in the manuscript. In introduction authors describe “AQP7 inhibitor Z433927330” which imply good selectivity, in the discussion they refer to Z433927330 as not specific “Our study highlights the poor pharmacologic properties of Auphen and Z433927330 and their impact on multiple AQPs. Both compounds are water insoluble compounds and are not AQP specific.”

I suggest that authors address this topic more rigorously to avoid any misunderstanding of the conclusions.

Author Response

Authors present their work on aquaporin inhibitors in treating breast cancer. This is an interesting, novel field of anticancer research and as such it is of interest to reader of Cancer.

 Methodologically, article is a standard work in this field, and as such appropriate. Conclusions are limited at this stage, authors clearly state the limitations of their study.

 Authors extensively studied the synergistic effects of Z433927330 and several anticancer drugs, this results will be of ineptest to all researchers in the field of anticancer drugs.

 My main concern is the selectivity profile of Z433927330, too often compound selectivity is not evaluated rigorously enough, therefore biological results are not so uniform as excepted.

Also authors have some issues with this topic in the manuscript. In introduction authors describe “AQP7 inhibitor Z433927330” which imply good selectivity, in the discussion they refer to Z433927330 as not specific “Our study highlights the poor pharmacologic properties of Auphen and Z433927330 and their impact on multiple AQPs. Both compounds are water insoluble compounds and are not AQP specific.”

I suggest that authors address this topic more rigorously to avoid any misunderstanding of the conclusions.

Response: We removed the description in the abstract that indicates that Z433927330 is AQP7 specific. In the introduction, we gave further clarification by including the IC50s.

Reviewer 3 Report

Comments and Suggestions for Authors

Reviewing the manuscript titled “Evaluation of the Mammalian Aquaporin Inhibitors Auphen 2 and Z433927330 in Treating Breast Cancer”, this study evaluates the efficacy of therapeutic inhibition of AQPs in breast cancer treatment using two AQP inhibitors, Auphen and AQP7 inhibitor Z433927330. The inhibitors affected expression of other AQP transcripts and proteins, demonstrating compensatory regulation between AQP family members. Auphen treatment extended overall survival but did not impact tumor burden. However, it made cells more responsive to chemotherapy or endocrine treatment, reducing overall AQP7 protein expression. The study highlights AQPs as therapeutic vulnerabilities in breast cancer metastasis.

In my opinion, the study is well organized, fills the research gap for the detection of anti-AQP7 inhibitors and I recommend it to be published. However, it would be even more enriching for the study to perform some in silico experiments showing how the proposed inhibitors bind to the AQP7 receptor.

Author Response

In my opinion, the study is well organized, fills the research gap for the detection of anti-AQP7 inhibitors and I recommend it to be published. However, it would be even more enriching for the study to perform some in silico experiments showing how the proposed inhibitors bind to the AQP7 receptor.

Response: We added a paragraph to the discussion that describes the in silico experiments with these inhibitors that already are published.